

# Local environmental context drives heterogeneity of early succession dynamics in alpine glacier forefields

Arthur Bayle[1][†][*], Bradley Z. Carlson[2][†], Anaïs Zimmer[3], Sophie Vallée[4], Antoine Rabatel[5], Edoardo Cremonese[6], Gianluca Filippa[6], Cédric Dentant[7], Christophe Randin[8], Andrea Mainetti[9], Erwan Roussel[10], Simon Gascoin[11], Dov Corenblit[10] & Philippe Choler[1]

[1] Univ. Grenoble Alpes, Univ. Savoie Mont Blanc, CNRS, LECA, F-38000 Grenoble, France

[2] Centre de Recherches sur les Écosystèmes d'Altitude (CREA), Observatoire du Mont-Blanc, F-74400 Chamonix, France

[3] Department of Geography and the Environment, The University of Texas at Austin, Austin, Texas, USA

[4] Conservatoire Botanique National Alpin (CBNA), F-73000 Chambéry, France

[5] Univ. Grenoble Alpes, CNRS, IRD, Grenoble-INP, Institut des Géosciences de l'Environnement (IGE, UMR 5001), F-38000 Grenoble, France

[6] Environmental Protection Agency of Aosta Valley, Climate Change Unit, Loc. La Maladière, 48, IT-11020 Saint Christophe (AO), Italy

[7] Parc National des Ecrins, Domaine de Charance, Gap, France

[8] Univ. Lausanne, Dept. of Ecology & Evolution / Interdisciplinary Centre for Mountain Research (CIRM), Biophore, CH-1015 Lausanne, Switzerland

[9] Biodiversity Service and Scientific Research, Gran Paradiso National Park, Torino, Italy

[10] Université Clermont Auvergne, CNRS, GEOLAB, Clermont-Ferrand, France

[11] CESBIO, Université de Toulouse, CNES/CNRS/IRD/INRAE/UPS, Toulouse, France

[†] These authors contributed equally to this work

*Correspondence to*: Arthur Bayle (arthur.bayle.env@gmail.com)

**Abstract.** Glacier forefields have long provided ecologists with a model to study patterns of plant succession following glacier retreat. While plant survey-based approaches applied along chronosequences provide invaluable information on plant communities, the "space-for-time" approach assumes environmental uniformity and equal ecological potential across sites and does not account for spatial variability in initial site conditions. Remote sensing provides a promising avenue for assessing plant colonisation dynamics using a so-called "real-time" approach. Here, we combined 36 years of Landsat imagery with extensive field sampling along chronosequences of deglaciation for eight glacier forefields in the south-western European Alps to investigate the heterogeneity of early plant succession dynamics. Based on the two complementary and independent approaches, we found strong variability in the time lag between deglaciation and colonisation by plants and in subsequent growth rates, and in the composition of early plant succession. All three parameters were highly dependent on the local environmental context, i.e., local vegetation surrounding the forefields and energy availability linked to temperature and snowmelt gradients. Potential geomorphological disturbance did not emerge as a strong predictor of succession parameters, perhaps due to insufficient spatial resolution of predictor variables. Notably, elapsed time since deglaciation showed no consistent relationship to plant assemblages, i.e., we did not identify a consistent order of successional species across forefields as a function of time. Overall, both approaches converged towards the conclusion that early plant succession is not stochastic as previous authors have suggested but rather deterministic. We discuss the importance of scale in deciphering the unique complexity of plant succession in glacier forefields and provide recommendations for improving botanical field surveys and using Landsat time series in glacier forefields systems. Our





work demonstrates complementarity between remote sensing and field-based approaches for both understanding and predicting future patterns of plant succession in glacier forefields.

## 1 Introduction

Glaciers in the European Alps began to retreat around the mid-19th century in response to changes in climate conditions driven first by shifts in precipitation (Vincent et al., 2005) and then by human-induced changes in aerosol concentrations in the atmosphere combined with warming (Painter et al., 2013, Sigl et al., 2018). Pronounced glacier retreat marked the end of the Little Ice Age (LIA), a multi-century period during which glacier terminal moraines were up to a few kilometers down valley from their current location (Matthews and Briffa, 2005, Gardent et al., 2014). As glaciers retreat, the

surface area of the glacier forefields (i.e., the area extending between the glacier snout and the moraine deposited during the LIA maximum extent) increases (Marta et al., 2021). Over the 20th century, the pace of glacier retreat in the Alps was not constant, with a few short glacier advances. However, this phase was recently followed by more consistent accelerating melting since the 1990s (Vincent et al., 2014), in response to warming air temperatures and associated reductions in snowpack depth and duration (Gobiet et al., 2014). In this context, recently deglaciated areas constitute particularly dynamic

ecosystems that are reshaping high mountain landscapes (Haeberli et al., 2017) and associated biodiversity and ecosystem services (Cauvy-Fraunie and Dangles, 2019, Ficetola et al., 2021). These ecosystems have been identified as hotspots of the widespread greening observed throughout the European Alps during recent decades (Bayle, 2020, Choler et al., 2021, Carlson et al., 2017; Dentant et al. 2022), calling for the need to better understand plant colonization dynamics in glacier forefields to predict future trajectories of alpine ecosystems (Huss et al., 2017).

Glacier forefields have long provided ecologists with a model to study patterns of plant succession along chronosequences of glacier retreat (Chapin et al., 1994), most often using a "space-for-time" approach (Pickett, 1989, Zimmer et al., 2018). This method uses the position of plant surveys to estimate time since deglaciation and relies on the assumption that within a glacier forefield, initial environmental conditions are consistent, and that pioneer species benefit from equal opportunity for establishment and growth over space and time (Johnson and Miyanishi, 2008). However, field

observations accompanied by a growing body of literature indicate that this approach is overly simplistic given the environmental heterogeneity of glacier forefields as well as the complexity of biological processes involved. Indeed, we now know that plant succession rates and trajectories are controlled by both abiotic and biotic processes, which in turn depend on regional landscape and local environmental factors, such as micro-climate (Joly and Brossard, 2007), substrate and disturbance regimes (Anthelme et al., 2021, Eichel et al., 2016), water and nutrient availability (Górniak et al., 2017), micro-

topography (Raffl et al., 2006, Scherrer and Körner, 2011) and broad scale gradients such as elevation and continentality (Garibotti et al., 2011, Rydgren et al., 2014, Robbins and Matthews, 2010, Robbins and Matthews, 2014). All these factors can lead to strong heterogeneity in vegetation dynamics within and between glacier forefields.

    Plant succession dynamics in glacier forefields, as elsewhere, can be broken down into three fundamental steps (Bradshaw, 1993): (i) diaspores reaching areas of bare ground, i.e., dispersal, (ii) successful and persistent establishment, and



(iii) plant succession, as community composition matures and develops over time. Wojcik et al. (2021) recently proposed a
novel conceptual model aimed at better understanding and predicting contrasting trajectories of plant succession dynamics in
glacier forefields based on the complex interplay between autogenic factors, i.e., time dependent biological succession, and
allogenic factors, i.e., external environmental factors such as climate or geomorphological disturbances. The authors suggest
that the importance of autogenic and allogenic components varies over time, with an initial stochastic phase (i.e., dispersal)
followed by a more deterministic phase defined by environmental factors, biological interactions, and bio-geomorphic
feedbacks (Eichel et al., 2016). Contrasting succession trajectories within glacier forefields are presented as the result of
variations in (i) time since glacier retreat; (ii) initial site conditions (heterogeneous micro-climate, substrate properties and
resource availability); and (iii) geomorphological disturbances (hillslope, torrential, periglacial, aeolian disturbances). In
addition to time, autogenic biological properties also shape plant succession dynamics: regional species pool composition
determines propagule pressure, while plant functional traits (wind dispersal and low seed mass) facilitate the arrival and
establishment of pioneer species (Franzén et al., 2019, Rosero et al., 2021, Schumann et al., 2016). At broader scales,
variability in plant succession rates between glacier forefields has been linked to elevation and continentality, which in turn
influence more direct environmental parameters such as temperature and snow cover duration (Robbins and Matthews,
2010).

In addition to assuming environmental uniformity and equal ecological potential over space, plant survey-based
approaches applied along chronosequences yield only a snapshot of plant community properties and fail to provide insights
into the temporal dynamics of succession. To address questions linking auto- and allogenic factors to time lags, i.e., time
between a surface deglaciation and its initial plant establishment, it appears necessary to adopt a "real-time" approach based
on annually resolved information on plant succession dynamics. While some studies have successfully implemented repeat
surveys of permanent plots in the context of glacier forefields (Fickert and Grüninger, 2018) repeat field surveys of plant
communities in often remote and hard-to-access mountain environments present major challenges in terms of cost and effort.

As a complementary approach to traditional plot-based surveys, which remain essential to understand eco-
geomorphological processes on the ground, remote sensing provides a promising avenue for assessing plant colonization
dynamics within and between glacier forefields using a so-called "real-time" approach. Beginning in 1984 with the Landsat
5 TM, followed by Landsat 7 ETM+ and Landsat 8 OLI sensors, Landsat satellites currently provide a 36-year archive of 30
m resolution images acquired at 16-day intervals over the globe's terrestrial areas. Availability of Landsat imagery since the
mid-1980s allows for investigating plant succession dynamics since the most recent observed advance of alpine glaciers and
in response to accelerating glacier retreat during recent decades. Vegetation indices such as the Normalized Difference
Vegetation Index (NDVI) provide a proxy of plant biomass (Tucker and Sellers, 1986), photosynthetic activity and
vegetation cover and have the potential to reliably quantify plant succession in rocky areas with low plant cover. As NDVI is
a non-physical and uni-dimensional quantity based on remotely measured reflectance, variations in quantities over time can
be disconnected from changes in plant cover on the ground, particularly in the context of heterogeneous topography (Bayle
et al., 2021). In addition to systematic errors caused by sensor limitations, variation in NDVI over time can be due to





atmospheric and cloud contamination (Masek et al., 2006), angular effects due to variation in sun-surface-sensor geometry or
topography (Nagol et al., 2015, Martín-Ortega et al., 2020), sensor degradation and calibration changes (Markham and
Helder, 2012) or between sensor spectral band pass (Steven et al., 2003). Overall, it has been shown that NDVI increases
near-linearly with fractional vegetation cover (horizontal density) until values reach between 80% and 90% at which point it
tends to saturate and increases slowly with increasing Leaf Area Index (vertical density). Remote sensing approaches have
already been used efficiently in the context of glacier forefields (Klaar et al., 2015, Fischer et al., 2019, Bayle, 2020,
Knoflach et al., 2021) and high sensitivity of Landsat-based NDVI to low plant cover has been demonstrated in glacier
forefields (Bayle et al., 2021) and in Antarctica (Fretwell et al., 2011).

Here, we investigated early plant succession dynamics (0-35 years since glacier retreat) in the context of eight
glaciers distributed across the southwestern European Alps. Specifically, we sought to answer the following ecological
questions: (1) Is observed heterogeneity in early plant succession dynamics indeed purely stochastic or can this variability be
linked to environmental factors? (2) Is early plant community composition consistent across sites and shaped by time
availability or heterogeneous and driven by local environmental context ? To address these questions, we utilized two
independent data sources: 36 years of Landsat imagery and 297 floristic field plots. First, we derived two indicators of
vegetation dynamics from Landsat time series, which are (i) time lag between ice melting and plant establishment, and (ii)
plant growth rate following establishment. Then, we investigated the spatial heterogeneity of these indicators and their
respective drivers using proxies of the local environmental and geomorphological context (allogenic factors). Second, we
assessed turnover in plant community composition between and within the eight glacier forefields using a "space-for-time"
approach and with regard to local environmental and geomorphological context (allogenic factors) and time since
deglaciation (autogenic factor). Finally, we questioned the capacity of field sampling to capture heterogeneity in vegetation
dynamics with respect to spatially exhaustive remote sensing approach, and we provide some recommendations to improve
field sampling methodology for future studies. In addition to testing theoretical expectations in glacier forefields across a
broad spectrum of environmental and ecological contexts, our study provides for the first time a clear roadmap for applying
widely available remote sensing data to quantify and improve our understanding of trajectories of plant colonization and
succession in glacier forefields.

## 2 Data and study site

### 2.1 Study site

Our study investigates eight glacier forefields distributed throughout the southwestern European Alps (Figure 1) in
France (Glacier Blanc, Saint-Sorlin, Gébroulaz, Tour and Pélerins), Switzerland (Orny) and Italy (Lavassey and Lauson).
Sites are distributed from 45° to 46° N and from 6° to 7° E across a variety of slopes and aspects, and elevations. Substrates
are highly variable both within and between forefields including exposed bedrock, chaotic blocks of various sizes, gravel,
and sand. Stream networks are often intricate, with strong seasonal and daily variability.



*2.2 Age of deglaciation*

Chronosequences of glacier outlines for the eight glaciers were initially obtained from various sources depending on the
country. For France (GB, GEB, STS, PEL, TR), glacier outlines were extracted from the GLIMS database (Gardent et al.
2014) which contain outlines dating from 1985/86 based on Landsat 5 TM (30m), 2003 based on Landsat 5 TM and 7 ETM+
(30m), 2006, 2008 or 2009 (depending on the glacier) based on sub-meter resolution images from BD ORTHO IGN and
finally 2014, 2016, 2018 (depending on the glacier) Spot 6/7 images (1.5 m). For Italy (LAU, LAV), glacier outlines for
1975 were obtained based on photo interpretation of Regional Technical Maps, 1999, 2005, 2012 and 2019 based on
orthophoto and Sentinel-2 (10 m). For Switzerland (OR), glacier outlines were extracted from the GLAMOS database
(Linsbauer et al., 2021). Overall, we were able to obtain chronosequences for the eight glaciers that approximately
corresponded to the Landsat time series historical depth. As our glacier contours database mixed automatic and manual
methods with sources at medium to high resolution, important quality differences were observed within and between glacier
chronosequence as older contours were mostly based on coarse resolution images through automatic approaches that tend to
perform poorly for debris-covered glaciers. To improve the consistency of our database and the reliability of further analysis,
we carried out manual photointerpretation of sub-meter historical images for all glacier contours. For STS, TR, OR and
LAV, small corrections were applied due only to the improvement of sources resolution, while for GEB, PEL and LAU,
which are totally or partly covered by debris, we substantially improved the delimitation by accounting for emissary streams,
crevasses and lateral screes reoriented by glacier movement. For GB, we also identified a large section of dead-ice that
detached from the main glacier in 2014, and which remains in 2021 (Bayle, 2020). We removed this area from further
analysis. Image sources are presented in Table S2 while detailed procedures to obtain our sub-meter glacier contours
database are presented for each glacier in Figures S1 to S6. Glacier outlines were manually delineated using ArcGIS
software (Esri, 10.4.1). A detailed description of the sources used to improve the glacier outlines dataset can be found in
Table S1.

We estimated continuous years of deglaciation (YOD) from glacier outlines using an interpolation method initially
designed for the creation of hydrologically corrected digital elevation models (DEMs). We used the Topo-to-raster function
in ArcGIS based on the ANUDEM program (Hutchinson et al., 2011), which is specifically designed to work with line
features as input, to obtain a raster indicating the YOD. Linear interpolation assumes that glaciers retreated at a constant rate
between two dated glacier extents. To evaluate this assumption, interpolated surfaces from Glacier Blanc were compared to a
denser chronosequence of deglaciation from Bayle (2020) that were not used in this work. The linear model between the
results from interpolation and ground-truth observation obtained a R² of 0.945 with a mean error of +/- 2 years (Figure S7).



*2.3 Vegetation field surveys*

Glacier forefields of GB, GEB, STS, PEL, TR, LAV, LAU and OR were surveyed in the months of July and August
in 2019 and 2020. For each forefield and chronosequence band (for example the zone deglaciated between 1983 to 2003), we
generated a set of 15 random sampling points, while ensuring a minimum distance of five meters between sampling points.
In the field we went as close as possible to these points using a GPS and excluded sites that were too dangerous to access
walking or that were under water or snow. For each plot, within a 2x2m quadrat we surveyed the percent cover of vascular
plants, mosses and lichens, biological soil crust (Khedim et al., 2021, Breen and Lévesque, 2008), and bare ground (which
was further subdivided into percent bedrock, sand, gravel < 2 cm, rocks between 2 and 20 cm and blocks larger than 20 cm
in diameter). We also recorded all vascular plant species as well as their relative cover and average vegetative height within
the quadrat. We had a total of 297 plots with 59, 37, 36, 38, 27, 20, 28 and 52 plots for GB, GEB, LAU, LAV, OR, PEL,
STS and TR respectively.

*2.4 Remote sensing data*

Landsat 5 TM, Landsat 7 ETM+ and Landsat 8 OLI standard level 1 Terrain-corrected (L1T) orthorectified images
from Collection 1 (geolocation error < 12 m) between 1984 and 2019 for 4 path/row (tiles) were downloaded from the
Landsat Earth Explorer data portal (http://earthexplorer.usgs.gov) at surface reflectance level of correction. Images with
average cloud cover < 80% only were selected as high cloud cover can reduce the number of available ground control points
and therefore the geolocation accuracy, and because cloud masking relies on clear-sky pixels to identify clouds. As a result, a
total of 2846 scenes were selected, 60% of which were from Landsat 5. To improve the robustness of our remote sensing
analysis, we applied state of the art methods to correct for inter-sensor spectral variation, Bidirectional Reflectance
Distribution Function (BRDF) and illumination related errors, and to mask cloud and cloud shadow cover in mountainous
context.

We applied the L7/L8 correction method described by Roy et al. (2016a) to our data to align L8 reflectance to L7.
No correction was applied on L5 or L7 as it was shown that the surface reflectance products from Landsat Ecosystem
Disturbance Adaptive Processing System (LEDAPS) are consistent through time, with no difference before and after the
2003 ETM+ Scan-Line Corrector (SLC) failure. The BRDF effects due to changes in solar and viewing zenith angle were
corrected using the c-factor approach (Roy et al., 2016b) based on the RossThick-LiSparse BRDF model (Schaaf et al.,
2002) and using an optimal normalised solar zenith angle defined by Zhang et al. (2016). We applied the Sun-Canopy-
Sensor + C model (Soenen et al., 2005) on the recommendation of Sola et al. (2016) to correct for illumination conditions
variation due to slope and aspect. Finally, clouds were masked for each scene using the MFmask 4.0 algorithm
(https://github.com/gersl/fmask) using the default DEM and a cloud probability threshold of 40%. This version has improved
clouds and cloud shadows detection by integrating auxiliary data, new cloud probabilities and novel spectral-contextual
features, which was crucial in mountainous area where integration of global DEM to normalise thermal and cirrus bands is





necessary (Qiu et al., 2017, Qiu et al., 2019a, Qiu et al., 2019b, Zhu and Woodcock, 2014, Zhu and Woodcock, 2012). Masks were computed for cloud probability threshold of 10%, 40% and 70% for a Landsat 7 ETM+ scene (195-029, captured the 16/07/2019) and visually compared to Fmask 3.3 (Figure S8). The probability of 40% offered good compromise between omission and commission errors and was thus selected. A detailed correction workflow is available in
supplementary materials.

To assess vegetation changes at pixel-scale over the eight glacier forefields, we computed the NDVI as follows:

$$NDVI = \frac{(R_{NIR} - R_{Red})}{(R_{NIR} + R_{Red})}$$

where $R_{NIR}$ and $R_{Red}$ are the Normalised BRDF-adjusted and topographically corrected reflectance in the NIR and Red bands, respectively. Then, we computed the NDVI annual maximum (NDVImax) available from 01 June to 31 August (day
of year ~ 152 to 243) from 1984 to 2019 to obtain a time series of an indicator of vegetation state at 30-m scale. Due to variations in cloud cover, image density and phenology between years, the max NDVI is not composed of the exact same day for each pixel. While the day of NDVImax could change due to phenological shifts throughout the time series, large changes should not be expected. Thus, we applied the same method as in Bayle et al. (2022). To prevent related errors in NDVI trend estimation, a year is discarded from the pixel time series if the mean day of year is superior/inferior to ±2σ of
the entire time series (Fig. S9).

**3. Statistical analysis**

Our data analysis workflow is based on using very high resolution chronosequence of deglaciation, Landsat time series and vegetation surveys to derive three "early succession dynamics indicators". The heterogeneity of these indicators
will be explored based on "predictor variables" divided in two categories: the local environmental context and potential geomorphological disturbances. These analyses intend to better understand the deterministic vs. stochastic nature of early succession dynamics in the theoretical framework proposed by Wojcik et al. (2020). Complete workflow is presented in Figure 2.

*3.1. Remote sensing-based indicators of succession dynamics*

Based on the NDVImax time series, we characterized the proglacial vegetation dynamics at Landsat pixel-scale using two indicators. The (i) Time Lag (TL$_{NT}$), i.e., the number of years between the Year of Deglaciation (YOD) and the year where the NDVI threshold (NT) is exceeded (YOE$_{NT}$), and the (ii) Growth Rate (GR$_{NT}$), i.e., the NDVImax trends computed from the YOE$_{NT}$ to the last year of the Landsat time series. Ideally, we would compute both indicators for a NT
that could identify the year of colonization by vegetation, but as NDVI tends to show noise unrelated to vegetation and that plant establishment is a progressive phenomenon occurring at small scale compared to pixel scale, such a threshold does not exist. Thus, we selected a NT based on Bayle et al. (2021). By comparing Landsat NDVI values to intra-pixel vegetation



cover derived from Unmanned Aerial Vehicle (UAV) image, they showed, for example, that a value of 0.071 efficiently discriminates pixels around 5% of vegetation cover (F-score > 0.75) with best efficiency with a value of 0.1 to discriminate

with more and less than 13% of vegetation cover (F-score > 0.85). Based on this work, we used an NT of 0.075 as it was a good compromise between specificity and sensitivity regarding plant cover.

Thus, we used $TL_{0.075}$ and $GR_{0.075}$ to quantify heterogeneity of intra and inter-glacier forefield succession dynamics. To consider a year as the one where the NT is exceeded, the NDVI of the two previous years must be lower, and the two next years to be higher. Theil-Sen trend estimator was applied for $GR_{0.075}$ estimation as it is resistant to outliers in short or

noisy series (Eastman et al., 2009). An example of an NDVI time series and all associated data is shown in Figure 3A. We evaluated our method by comparing plant cover (%) obtained from field sampling between plots identified as vegetated or unvegetated. Finally, as the Landsat time series is limited to the last 40 years, there is a bias in the $TL_{0.075}$ value as it is directly constrained by the YOD. For example, a pixel deglaciated in 2010 could only be colonized in the 9 following years, thus limiting the absolute value of $TL_{0.075}$ between 1 and 9 years. To bypass this bias, we computed the anomalies of $TL_{0.075}$

and $GR_{0.075}$ as a function of YOD, which is a more relevant measure of heterogeneity in succession dynamics across the eight glacier forefields (Fig. 3B and 2C).

Finally, we implemented two random forest classification analyses to assess relationships between anomalies of $TL_{0.075}$ and $GR_{0.075}$ and predictors (Breiman, 2001). We classified the two indicators into three categories: positive anomalies (anomalies of $TL_{0.075}$ | $GR_{0.075}$ > 0), negative anomalies (anomalies of $TL_{0.075}$ | $GR_{0.075}$ < 0) and no vegetation detected. For

TL, it resulted in 337, 374 and 1977 samples for positive, negative and no detection, respectively, and for GR, 441, 288 and 1959 samples for positive, negative and no detection, respectively. Only 337 and 288 samples were conserved for TL and GR, respectively, to equalize the sample size of each class.

Predictors variables included (i) elevation, (ii) local vegetation and (iii) snow-free growing degree days (SF-GDD) to represent the environmental context (allogenic factors) of the glacier forefields. Elevation was obtained from the 25-m

resolution European Digital Elevation Model (EU-DEM, version 1.1; https://land.copernicus.eu/imagery-in-situ/eu-dem/). Snow-free growing degree days (SF-GDD) maps were calculated through the combined use of the Snow-melt out date (SMOD) product at 20-m resolution derived from Sentinel-2 time series analysis (Gascoin et al., 2019, Barrou Dumont et al., 2021), and the SAFRAN-CROCUS climatological model from Météo-France (Vernay et al., 2022), which provides the average daily temperature for each French massif and for 300 m elevation bands (for Swiss and Italian sites, we applied data

from the French Mont-Blanc and Vanoise massifs, respectively). To obtain the SF-GDD for each plot, we computed the cumulative sum of daily average air temperature above 0°C between snow melt-out date and August 1 for the year 2019. SF-GDD is representative of the heat accumulated by vegetation during the growing season and is known to be a key variable for habitat distribution and alpine plant community properties (Choler, 2015, Choler, 2018, Carlson et al., 2015). We defined local vegetation surrounding the glacier forefield as the expected local vegetation productivity (NDVImax) for a given

elevation outside of the glacier forefield, which we considered to be a proxy of the vegetation proximity and type. To




compute this indicator for each plot, we averaged the NDVImax of the year of deglaciation for the 100-m elevational bands of the plot within a radius of 500-m and by excluding pixels within the glacier forefield.

We also calculated (i) Flow accumulation and (ii) LS-factor (slope length and steepness-Factor) to represent potential geomorphological disturbances (allogenic factors). The LS factor was derived from the EU-DEM and computed using the original equation proposed by Desmet and Govers (1996). It combined the S-factor, which accounts for slope angle, and the L-factor that defines slope length. The combined LS-factor describes the effect of topography on soil erosion and thus is a proxy of "potential" instability due to gravity-related processes. Flow accumulation was also derived from the EU-DEM and computed using a Multiple Flow Direction algorithm. Both the LS-factor and Flow accumulation were computed in SAGA (Conrad et al., 2015).

We screened highly correlated variables ($r^2 > |0.7|$) by computing pairwise correlations. We then removed elevation as it was highly correlated to SF-GDD ($r^2=-0.73$) and Local vegetation ($r^2=-0.84$). Next, we randomly partitioned the data set into sets for model training (two-thirds) and evaluation (one-third), and then fit random forest models to optimize out-of-bag classification accuracy. We reassessed the classification accuracy using the data withheld for model evaluation. We repeated this operation 100 times. Lastly, we computed predictor importance using the mean decrease in accuracy metric. Predictor importance was calculated using a permutation-based importance measure where one measures the effect of reshuffling each predictor on model accuracy. Finally, we generated partial dependence plots for the first variable in terms of importance as defined above to assess how class-specific classification probabilities varied across the range of the predictor while holding all other predictors at their average value. The distribution of predictors was constrained to regions with enough data given that partial dependence plots tend to overinterpret regions with few observations. We used the random Forest, caret, and pdp R packages to implement random forest models and to evaluate their performance (Liaw and Wiener, 2002, Greenwell, 2017).

### 3.2. Heterogeneity of succession dynamics and drivers

To model succession dynamics from the floristic data, we used Non-metric Multidimensional Scaling (NMDS) to perform an unconstrained ordination of the plot by species table. NMDS is a rank-order based multivariate technique that is particularly robust to identify a few important axes of floristic variations in community composition data (Minchin, 1987). We first discarded species with less than five occurrences to limit the proportion of sites with no shared species, as this may complicate the ordination. The resulting table included a total of 297 plots and 119 species. Absolute species covers were transformed into relative species covers using the Wisconsin standardisation, where species covers are first standardised by maxima and then site covers by maxima. Finally, the species covers were square rooted. These transformations are commonly found to improve the results of the NMDS (Legendre and Gallagher, 2001). We computed a distance matrix using the Bray-Curtis dissimilarity index on transformed species cover. To avoid local minima, we performed several NMDS with random starts and selected the solution with the minimum stress, i.e., the extent to which the distance between sites in the




specified number of dimensions differs from original distances. We used a vector fitting approach to test linear relationships
between NMDS site scores and three sets of variables corresponding to time since deglaciation (autogenic factor), and
allogenic factors including environmental (local vegetation, SFGDD) and geomorphological context (LS-factor and Flow
accumulation). Analysis was performed using the metaMDS and envfit functions of the R package vegan (Oksanen et al.,
2020).

In addition to environmental context and potential disturbances regime variables (allogenic factors) presented in
section 3.1, we also considered sediment granulometry as measured in the field. It was obtained by performing a cumulative
weighted mean for each plot of grain size (sand < 0.5 cm, gravel < 2 cm, cobbles between 2 and 20 cm and boulders between
20 cm and 1 m in diameter) visually estimated for floristic plots, by assigning the median value for each class and calculating
a cumulative weighted mean. Finally, we derived the time since deglaciation by subtracting the year of deglaciation obtained
as explained in section 2.2 by the year of field sampling.


## 4. Results

### 4.1. Vegetation dynamics heterogeneity assessed through remote sensing

Using 36 years of NDVImax obtained from the Landsat time series, we found strong heterogeneity within and
between glacier forefields. It took on average 10 years following deglaciation to reach an NDVI value of 0.075 for GB in
comparison with for example, 27 years for GEB, 17 for TR, 25 for STS, or even no vegetation detected for LAU over the
entire period (Fig. 4). Within glacier forefields, plant colonisation rates varied depending on the glacier with GB depicting
the highest heterogeneity in NDVI dynamics following deglaciation, and TR being the more homogenous with almost no
pixel with an NDVI value under 0.075 after 30 years (Fig. 4). Also, we did not find vegetated pixels for LAU over the study
period, which was corroborated by low plant cover recorded in the 2019 field survey (on average 4%). By simply
considering pixels as colonised by plants or not, we found that after 30 years since deglaciation 80% of GB, TR, and STS
forefields were colonised, and only 30% for PEL and LAV (Fig. 5). Overall, considering anomalies of TL0.075, we found
similar results with GB being the most dynamic forefield with pixels that tended to become colonised by vegetation three
years faster and with higher growth rates than average (Fig. 6A).

The two random forest models achieved an overall accuracy of 75% and 72% for TL and GR respectively, which
translated a good capacity of the four predictors to classify time lag and growth rate anomalies as positive, negative, or
absent. Mean decrease in accuracy (MDA) was used to order the four predictors according to their importance of overall
classification accuracy (Fig. 7). SF-GDD was overall the most important predictor for both models (MDA = 0.128 [0.127]
for TL [GR]) compared to Local vegetation, LS-factor, and Flow accumulation (respectively, 0.062 [0.052], 0.051 [0.041]
and 0.044 [0.04] for TL [GR]). Overall, allogenic factors describing environmental context were more important than
potential geomorphological disturbances variables (Fig. 7A and 7B). Classification probabilities showed that faster
colonisation occurred with SF-GDD > 900 (Prob > 0.5) while slower colonisation occurred mostly between 500 and 900 SF-





GDD. Slower growth rates occurred with SF-GDD > 500 while faster growth rate occurred sporadically above 900 SF-GDD (Fig. 7D). The class of undetected vegetation occurs mostly under 500 SF-GDD for both TL and GR (Fig. 7C and 7D).


### 4.2. Floristic plots are representative of glacier forefields vegetation dynamics

By comparing plant cover of the 297 floristic plots to the detection of vegetation as detected by Landsat NDVI, we found that the 221 floristic plots identified as unvegetated had an average plant cover of 5% (Fig. 6C). These results matched the initial sensitivity targeted considering the scale difference between Landsat pixel and floristic plots. Similarly, we found

anomalies of TL0.075 and GR0.075 to be representative of the overall vegetation dynamics of glacier forefields as values tended to be similar when we compared only pixels that overlapped floristic plots to the entire forefield (Fig. 6A and 6B).

### 4.3. Environmental context drives plant species assemblages and succession dynamics

The first axis of the NMDS showed a thermal floristic gradient representative of species turnover along an elevation

gradient throughout the southwestern European Alps (Fig. 8A and 8B). Cold-adapted high alpine specialist species, hereafter referred to as cold successional species, included low-stature, pioneer hemicryptophytes of sparsely vegetated screes, talus, and rock debris such as *Linaria alpina*, *Cerastium uniflorum* and *Saxifraga spp*. More thermophilous species typically found at lower elevations, hereafter referred to as warm successional species, included phanerophytes (*Picea abies*) and chamaephytes (*Vaccinium uliginosum*). Notably, we observed warm successional species as pioneer species in the context of

the Glacier Blanc, Pèlerins, and Tour glaciers, including tree and shrub species such as *Larix decidua* and *Salix laggeri*.

NMDS1 was positively correlated to local vegetation (r²=0.6290), SF-GDD (r²=0.7387) and negatively correlated to elevation (r²=0.6747). It was poorly correlated to other factors such as the LS factor (r²=0.2621), debris size (r²=0.0836), Flow accumulation (r²=0.0407) and time since deglaciation (r²=0.0046). By comparing NMDS1 scores to time since deglaciation, we found that elapsed time following deglaciation led to more late successional species for all glacier forefields

(except PEL) but at a rate insufficient to surpass the effect of initial plant community composition (Fig. 9). We found that the initial starting point of the succession to be mostly driven by allogenic factors describing the environmental context (average r²=0.68), and not potential geomorphological disturbances (average r²=0.1288) or time available (r²=0.0046) (Table 1). Finally, we found that the heterogeneity of vegetation dynamics described through the anomalies of TL0.075 and GR0.075 were linked to the NMDS1 scores with slower colonisation (positive TL0.075 anomalies) and growth (negative GR0.075

anomalies) rate corresponding to lower NMDS1 site scores.

## 5. Discussion

Our comparative study of vegetation dynamics in glacier forefields based on two complementary and independent approaches provides insight into the heterogeneity of early plant succession dynamics after glacier retreat at the regional





scale. First, using the Landsat time series, we found strong variability in the time lag between deglaciation and colonisation by plants and plant growth rate within and between glacier forefields (Fig. 4, 5 and 6). We showed that this heterogeneity was mostly explained by the local environmental context, i.e., local vegetation surrounding the forefields and energy availability linked to temperature and snowmelt gradients, rather than potential geomorphological context (Fig. 7). Similarly, by analysing turnover in plant species assemblages derived from field sampling, we found that the composition of early plant

succession communities established on glacier forefields, i.e., initial starting point of plant succession, was also strongly correlated to allogenic factors describing the environmental context rather than potential geomorphological disturbances (Fig. 8 and Tab. 1). In most cases, the increased time since deglaciation was found to be insufficient to overstep the determinism of the initial starting point, highlighting the importance of the local environmental context to understand plant succession in glacier forefields (Fig. 9). Overall, both approaches converged towards this conclusion (Fig. 10) suggesting

that early plant succession is not stochastic as previous authors have suggested (Wojcik et al., 2021) but rather deterministic.

### 5.1. Succession dynamics in glacier forefields are shaped by local environmental context

In our analysis, early vegetation dynamics in glacier forefields can be divided into two phases: (i) the lag between deglaciation and plant establishment; and (ii) the succession dynamics following plant establishment (Fig. 2). We analyzed

the heterogeneity of the first phase through two approaches based on remote sensing. First, as we distinguished vegetated from non-vegetated pixels, we were able to estimate the proportion of glacier forefields to become colonized by vegetation regarding time since deglaciation. We found strong heterogeneity between glacier forefields, with 80% of the forefield colonized by plants after 30 years for GB, TR, and STS, while the five other forefields did not reach 50% of vegetation cover over the same period, with no vegetation identified for LAU (Fig. 5). The second approach directly quantifies the time lag

between deglaciation and plant establishment. Nonetheless, as the range of absolute time lag is constrained by the length of the period between the year of deglaciation and the end of the time series, time lag could only be compared as anomalies regarding the median time lag for each year of deglaciation (Fig. 7A).

Time lag between deglaciation and plant establishment has been shown to be dependent on the proximity and availability of seed sources (Erschbamer et al., 2001, Stöcklin and Bäumler, 1996, Tackenberg and Stöcklin, 2008) and

species ability to disperse (Fickert and Grüninger, 2018), findings which are consistent with our results. Indeed, using partial dependency analysis, we showed that the time lag anomalies (including the absence of vegetation detected) are distributed along the SF-GDD gradient, which in turn is highly correlated to both elevation and local vegetation. These results confirm that high elevation forefields surrounded by sparsely vegetated scree slopes in the immediate surroundings tend to be colonized more slowly than forefields at lower elevation with dense patches of vegetation nearby. This conclusion is

supported if we consider the most and least dynamic forefields, respectively GB and LAU, which match these characteristics. In terms of spatial distribution, the GB forefield is also located at lower latitude and is the only south facing glacier forefield among the eight glaciers studied (Fig. 1), both of which contribute to earlier snowmelt-out and greater



accumulation of growing degree days for an equivalent elevation located on a north-facing glacier situated at a higher latitude. Indeed, the left bank of the GB forefield has been described by Bayle (2020) as highly dynamic, which is confirmed here with colonization occurring within 1 to 5 years after deglaciation, as reported elsewhere in the European Alps (Burga et al., 2010, Cannone et al., 2007, Fickert and Grüninger, 2018). This specificity is known to be due to the proximity of a dense vegetation patch which was located at less than 100 m to the glacier tongue in 1984 and at low elevation (2400 m a.s.l.). Conversely, the LAU forefield deglaciated at the same period is located at 3100 m a.s.l., further to the north in the Grand Paradiso National Park with only sparse vegetation nearby (Mainetti et al., 2021).

### 5.2. The importance of scale for understanding drivers of plant succession dynamics

While we did not observe high explanatory power of geomorphic variables in explaining the heterogeneity of succession dynamics in glacier forefields, a large number of studies have shown otherwise (Gurnell et al., 2001, Moreau et al., 2008, Burga et al., 2010, Eichel et al., 2013, Temme and Lange, 2014, Klaar et al., 2015, Heckmann et al., 2016, Eichel et al., 2018, Eichel, 2019, Miller and Lane, 2018, Wojcik et al., 2020, Wojcik et al., 2021). Biogeomorphological studies emphasise that landscape dynamics within glacier forefields depend on the balance between stabilising and destructive forces (Eichel et al., 2018). Indeed, proglacial plant succession in the wake of deglaciation alters site conditions and decreases the magnitude and/or the spatial extent of geomorphological disturbances (Gurnell et al., 2000, Moreau et al., 2008, Eichel, 2019, Miller and Lane, 2018). More specifically, the biogeomorphic phase, characterized by feedbacks between abiotic and biotic processes, is a key stage of landscape stabilisation in glacier forefields. However, exacerbated fluvioglacial and hillslopes processes within the deglaciated area during the first stage of the so-called paraglacial period may also delay the succession rate through rejuvenation of proglacial deposits (Wojcik et al., 2020).

Through the remote sensing and floristic approaches used here, GB displayed exceptionally fast colonisation in certain areas but also high heterogeneity within the forefield, with up to 30 years of difference in time lag for the same year of deglaciation (Fig. 6A). We attribute this heterogeneity to spatial variability in the intensity of potential geomorphological disturbance (Lardeux et al., 2015). Specifically, a gullying area on the right bank of the forefield remains active and thus prevents vegetation establishment, as described by Bayle (2020) and Lardeux et al. (2015). This geomorphic activity is known to be the result of an active eroding slope supported by a slowly melting ice-cored moraine, which is a phenomenon too fine and complex to be captured by our low-resolution variables of potential instability. This example shows that, although the sum of degree days during the growing season (SF-GDD) and local vegetation explained most of the observed heterogeneity in observed vegetation dynamics (lag and growth rate), finer variables estimating direct instability and geomorphic activities based on a high-resolution imagery or field measures would locally change the balance of importance and improve the predictive capacity of models (Fig. 7). The PEL glacier provides a further example, where we observed long lag times and slow growth rates despite relatively low elevation and accordingly high SF-GDD, which we attribute to the extremely chaotic and blocky substrate observed in the field. Although we did attempt to capture potential geomorphological



disturbances through substrate heterogeneity by calculating the weighted mean estimate of block size for floristic plots, we found similar results in this case compared to coarser DEM-based variables. Thus, we lacked a spatially continuous and ecologically relevant estimate of substrate properties. Estimating block size and geomorphological properties using a remote sensing approach has been explored (Westoby et al., 2017, Vázquez-Tarrío et al., 2017, Langhammer et al., 2019, Eichel et

al., 2020, Lang et al., 2021) and constitutes an important perspective for enhancing the analysis conducted in the present study, perhaps especially in regard to later plant succession dynamics known to be particularly linked to bio-geomorphic feedbacks (Eichel et al., 2016, Miller and Lane, 2018, Lane et al., 2016).

Furthermore, the poor predictive capacity of our potential geomorphological disturbance variables could be explained by scale differences. Most studies highlighting the importance of geomorphic activities in explaining succession

dynamics heterogeneity have been conducted at the glacier forefield scale at sub-meter spatial resolution. Our regional-scale approach showed that the local environmental context drives an initial starting point and that other local factors are insufficiently determinant to overrule it (Fig. 9A). We hypothesise that for our analysis, the importance of geomorphological variables and processes were overshadowed by broader-scale and more contextual drivers such as energy availability.

*5.3. The Landsat time series effectively captures vegetation dynamics in a real-time approach*

The Landsat time series offers the possibility to study vegetation dynamics using a real-time approach, given its temporal resolution and depth (Wulder et al., 2019, Bayle et al., 2021). For the first time, we used the Landsat archive since the mid-1980s to quantify real-time plant colonization dynamics at the scale of glacier forefields, based on ecologically relevant indicators of time lag and growth rate following establishment. In the context of the observed generalized greening

of the European Alps (Choler et al. 2021), these parameters provide further insight into trajectories of plant colonization in the context of peri-glacial greening hotspots. Nonetheless, this approach inevitably comes with some uncertainty and bias that merit discussion.

Relationships between NDVI and certain biophysical properties of vegetation canopies, such as leaf area index (LAI), vegetation cover and biomass have been widely studied (Ormsby et al., 1987, Wittich and Hansing, 1995), including

in the context of glacier forefields (Bayle et al., 2021, Knoflach et al., 2021) and in sparsely vegetated sites in the Antarctic (Fretwell et al., 2011). Further testing of relationships between NDVI and plant canopy properties in the context of glacier forefields remains necessary, however, to better understand the effects of specific conditions unrelated to vegetation that can alter reflectance, such as low organic content in soil (Todd et al., 1998), coarse granulometry and complex angular effects due to micro-topography (Bayle et al., 2021). Despite the difference in area between field plots (2x2m) and Landsat pixels

(30x30m), we found similar sensitivity between ground truth vegetation cover and NDVI values to that of Bayle et al. (2021) and thus validate the within this study region  an NDVI value of 0.075 efficiently identifies pixels with more of less than 5-10% of vegetation cover (Figure S10). Additionally, we found that to take full advantage of the time series, the presence of clouds was problematic as the actual cloud mask applied in the data distributed by the USGS (Fmask 3.3) is inefficient in



high elevation areas as soil temperature is too low and is often confused with clouds (Qiu et al., 2017, Qiu et al., 2019b).
Thus, we recommend using the modified MFmask 4.0 to improve the number of images available (Figure S8). Despite these challenges, our study thus confirms that the Landsat time series can be efficiently used to monitor vegetation cover changes over time in the context of glacier forefields.

*5.4. Field sampling recommendations*

Several studies have shown that the initial site conditions defined by substrate material, topography, micro-climate, landscape surroundings, as well as by erratic frequencies and/or magnitudes of natural disturbances between sites, are of high importance in determining plant succession dynamics in glacier forefields (Joly and Brossard, 2007, Walker and Wardle, 2014, Wojcik et al., 2020, Eichel et al., 2016). Repeated visits of permanent plots represent an alternative to the regular space-for-time approach as they provide a more informative and reliable measure of succession. This approach is
costly in terms of effort, however, and furthermore it can be difficult to mark permanent plots in unstable terrain often found in glacier forefields (but see Bakker et al., 1996). An intermediate and less-costly approach consists in measuring variations in initial site conditions and geomorphological disturbances along the chronosequence, as proposed by Wojcik et al. (2021). In our field campaign, the eight glacier forefields were surveyed based on a random sampling approach along the chronosequence of deglaciation, however vegetation plots captured overall heterogeneity within forefields with varying
degrees of success (Fig. 6). In accessible glacier forefields of small size and limited geomorphic activities (TR, PEL, GEB, STS and OR), our approach worked as floristic plots were found to be representative of the vegetation dynamics as assessed from the spatially exhaustive remote sensing approach (Fig. 6). The representativity was less evident for LAV, which can be explained by the large size of the forefield and the presence of cliffs and a lake that drastically constrained accessibility. Noticeably, our results show that the within forefield variability in vegetation dynamics can be equivalent to the variability
observed at regional scale, thus leading to massive bias in our capacity to extrapolate results when studies rely solely on a space-for-time approach applied to a single glacier forefield.

      Our study provides lessons that could contribute toward improving future studies carried out in glacier forefields. Although explaining the absence of vegetation was not our question here, to understand where plants will establish, we recommend sampling points characterised by an absence of vegetation in addition to vegetated areas. Indeed, the first step in
the primary succession dynamics is whether vegetation can colonise a surface. Thus, the absence of vegetation on deglaciated surfaces should be considered as an extreme case of equal ecological relevance as vegetated plots, especially in the context of biogeomorphic feedbacks and regular disturbances minimising the probability of establishment and germination. Thus, in the perspective of an exhaustive and representative sampling of the glacier forefield, collecting information on the absence of vegetation appears to be crucial for future field campaigns. Overall, we recommend the
following procedure for future field campaigns:

(1)       Randomly select a predefined number of points using GIS software within targeted glacier forefields.



(2)      Collect information unrelated to vegetation on plots whether there is vegetation or not and keep track of unreachable random points due to dangerous access as it is a marker of high geomorphic activity.

(3)      Measure plant community structure and composition, including functional traits, both within and outside of the glacier forefields to capture the local environmental context through field sampling.

## 6. Conclusion

We quantified heterogeneity in plant succession dynamics based on 36 years of optical satellite imagery and 297 floristic plots distributed among eight glacier forefields in the southwestern European Alps. The projection of autogenic and allogenic factors according to the definition presented in Wojcik et al. (2021) shows that the pioneer plant community composition is strongly correlated with environmental context rather than time since deglaciation. Time since deglaciation is typically identified as the main driver of succession dynamics, as it is fundamentally intrinsic to the idea of succession. Nonetheless, whether time since deglaciation is identified as the main driver of succession is a matter of scale. We showed that, indeed, if we consider each forefield independently, a clear successional gradient emerges as time since deglaciation increases. But when considering all eight glacier forefields together, energy availability and initial species composition emerged as the key parameters shaping successional dynamics. In the case of both remotely sensed vegetation indices and plant field surveys, we found that this initial starting point was strongly correlated to the local environmental context, rather than the geomorphological context (Table 1). Overall, our findings suggest that early stages of plant succession in glacier forefields in the European Alps are highly dependent on the local environmental context and less stochastic than previous studies have suggested.

In the conceptual framework of Wojcik et al. (2021), our findings suggest a reinterpretation of the importance of local environmental context in the initial stages of primary succession, which is considered to be highly stochastic (Chase and Myers, 2011, Dini-Andreote et al., 2015, del Moral, 2009, Mong and Vetaas, 2006, Marteinsdóttir et al., 2010). In contrast, we found that this initial phase was driven by plant opportunism originating from the local species pool, which is a function of environmental context and other biogeographic factors. We highlighted that during 30 years of succession, time since deglaciation became a meaningless predictor of plant community composition when multiple and highly contrasting glacier forefields were considered, thus pointing to the importance of quantifying more direct drivers of succession dynamics including both environmental and biological factors. Our work highlights the ongoing need for process-based studies combining remote sensing and field techniques to improve our understanding of local heterogeneity in plant colonisation trajectories, and furthermore provides a promising basis for predicting future trajectories of plant succession in the wake of ongoing glacier retreat during the coming decades using widely available remotely sensed predictors.

## Data availability



Research data can be accessed upon request to the corresponding author. Part of the data used in this paper remains under
exclusivity as it was obtained through multiple programs and partnerships.

**Author contribution**

**Conceptualisation** AB and BC **Data curation** All authors contributed **Formal analysis** AB and BC **Writing – Original
draft and preparation** AB and BC **Writing – review and editing** All authors contributed


**Competing interests**

The authors declare no competing interests.

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





**Figure 1.** Distribution of the 8 glacier margins in France, Switzerland, and Italy with corresponding abbreviations. Floristic plots are indicated in yellow points while glacier outlines are shown in thick coloured lines. Colours do not indicate similar outlines date but chronosequence of deglaciation. Projection and sources details are indicated in the bottom right panel.





**Figure 2.** Complete analysis workflow showed by methodological sections. Early succession dynamics indicators are shown in green while predictors variables are shown in red and brown.





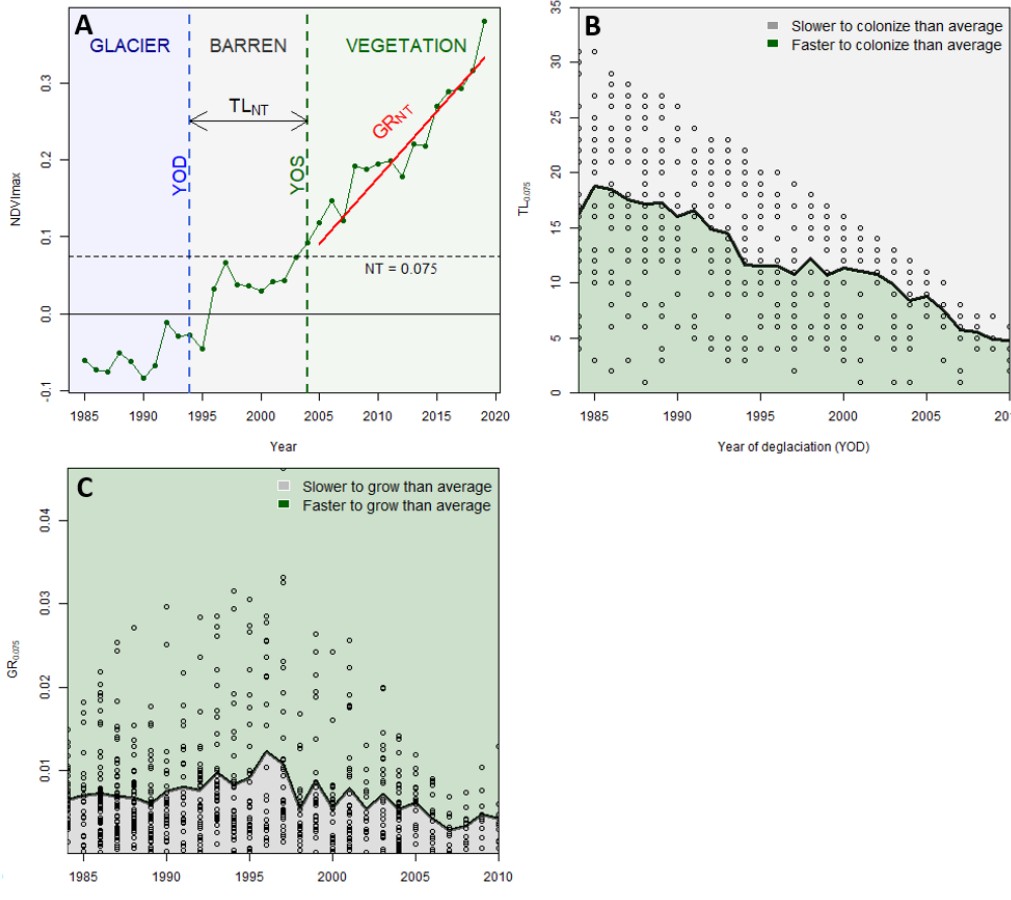

**Figure 3.** (A) Example of indicators derived from the Landsat-based NDVImax time series and deglaciation data for latitude: 44.932938 and longitude: 6.409681. (B) Time Lag (TL) and (C) Growth Rate (GR) for an NDVI threshold of 0.71 for the 8 margins according to year of deglaciation. Thick black lines represent the median value for each year of deglaciation used to compute anomalies. Blue shaded areas represent faster colonization and growth rate than average, in opposition to red shaded areas.






**Figure 4.** Distribution of NDVImax values according to the number of years since deglaciation for the 8 glacier margins. Error bars indicate standard deviation and numbers indicate the numbers of pixels for a given number of years since





deglaciation. Horizontal dashed lines show an NDVI of 0.075. Numbers of pixels for each year since deglaciation are indicated above each error bar.

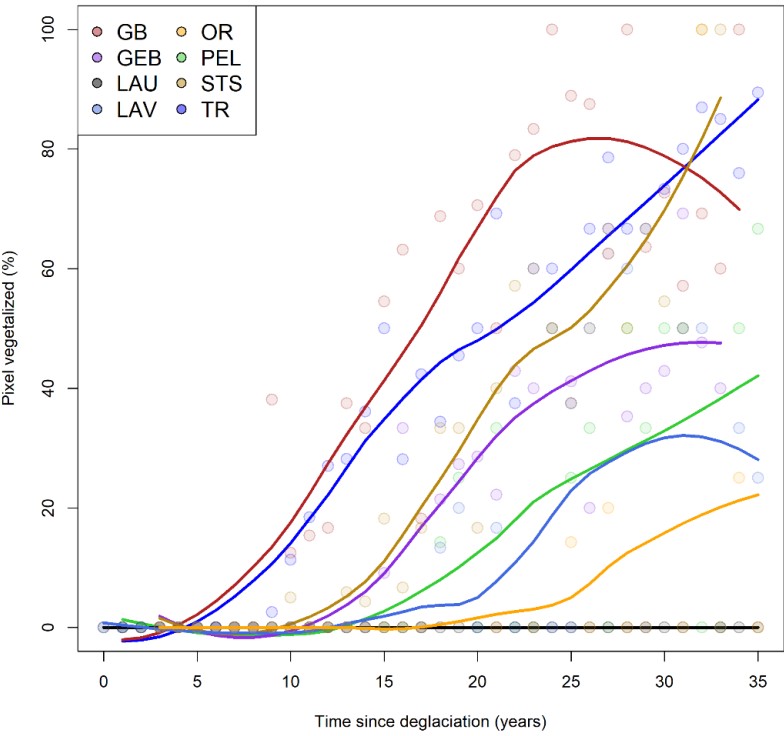


**Figure 5**. Proportion of pixels identified as vegetated according to time since deglaciation for all pixels within the 8 glacier forefields. Thick lines are computed from loess function with span = 0.7.



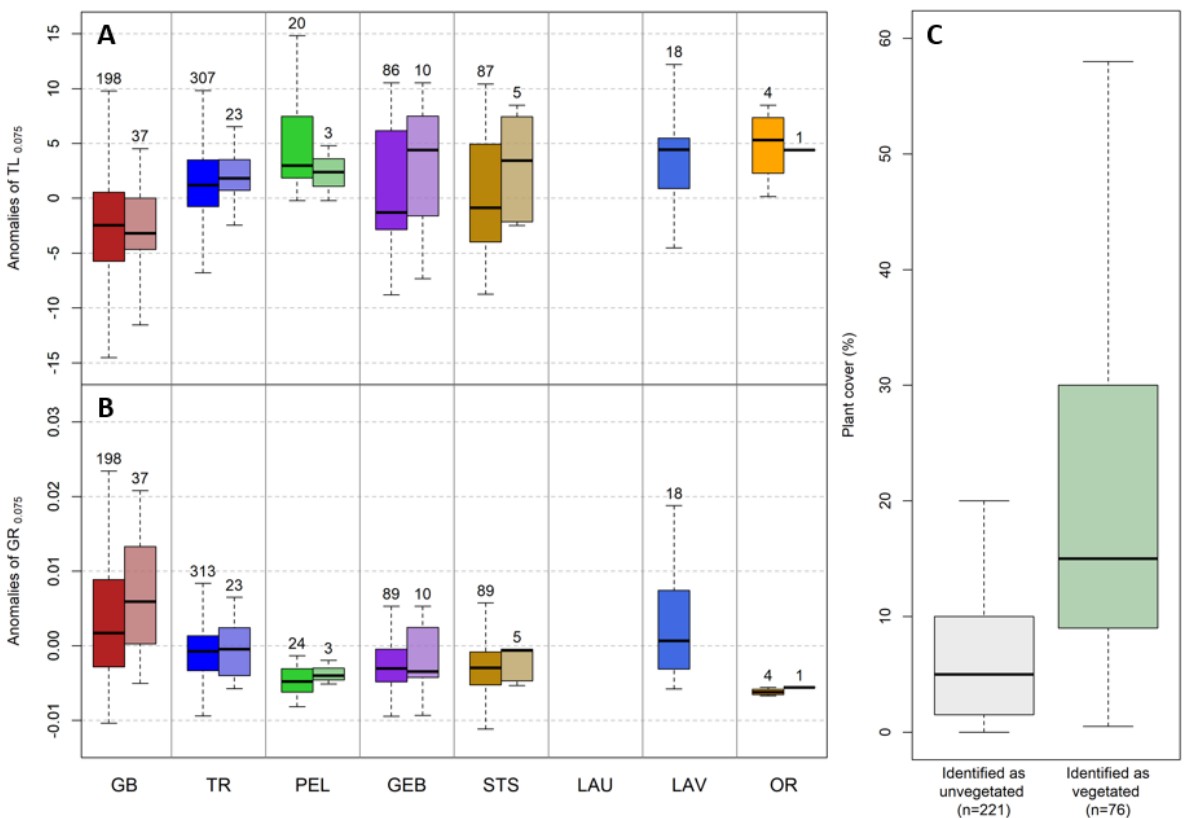

**Figure 6.** Distribution of (A) Time Lag and (B) Growth Rate anomalies for the 8 glacier margins for an NDVI threshold of 0.075. For each glacier, the left boxplot with full colours corresponds to all pixels while the right boxplot with shaded colours corresponds only to pixel overlapping floristic plots. (C) Distribution of plant cover (%) for floristic plots that has been identified as vegetated/unvegetated based on the remote sensing approach.



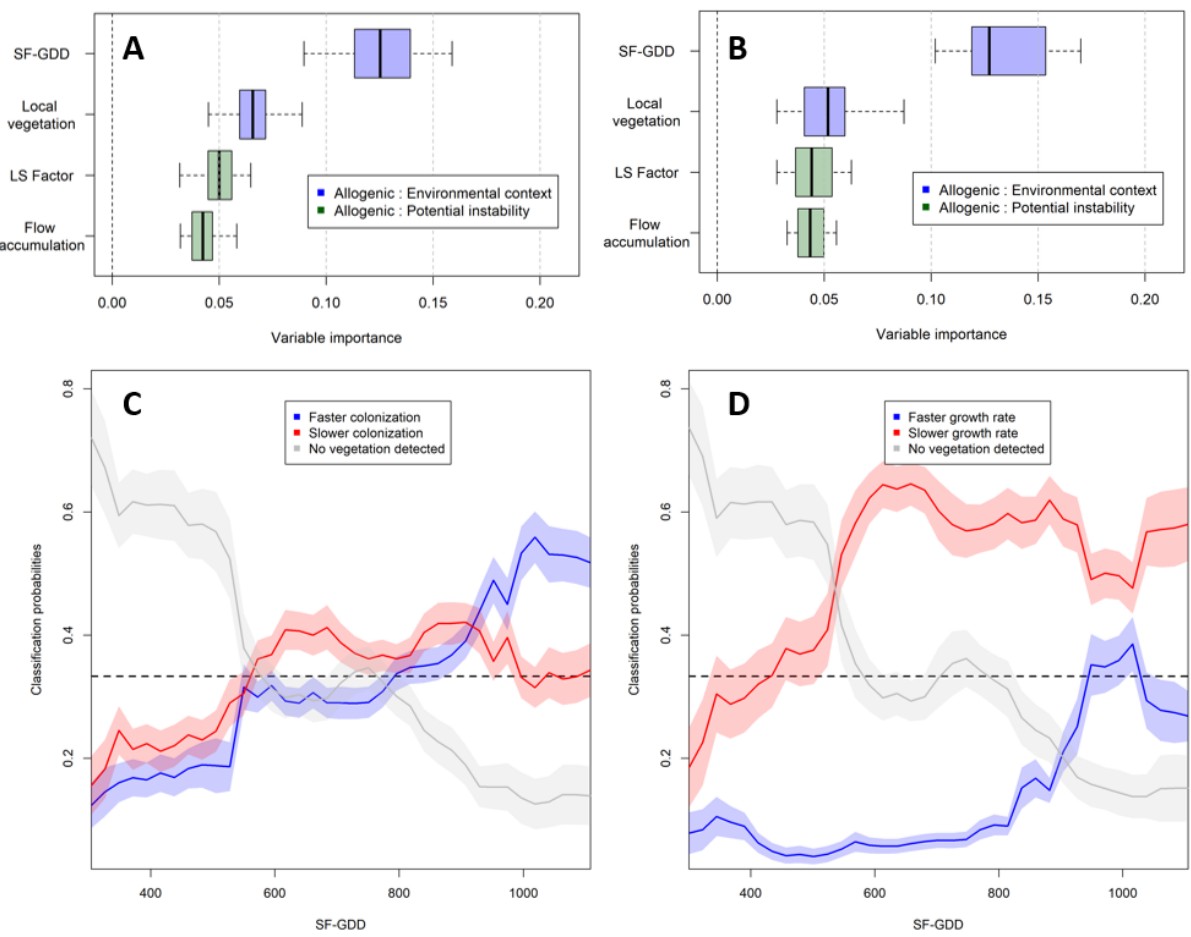

**Figure 7.** Variable importance from random forest model considering the three classes of (A) TL and (B) GR. Partial
dependency plot illustrating how SF-GDD affects class probability for the three classes of (C) TL and (D) GR.



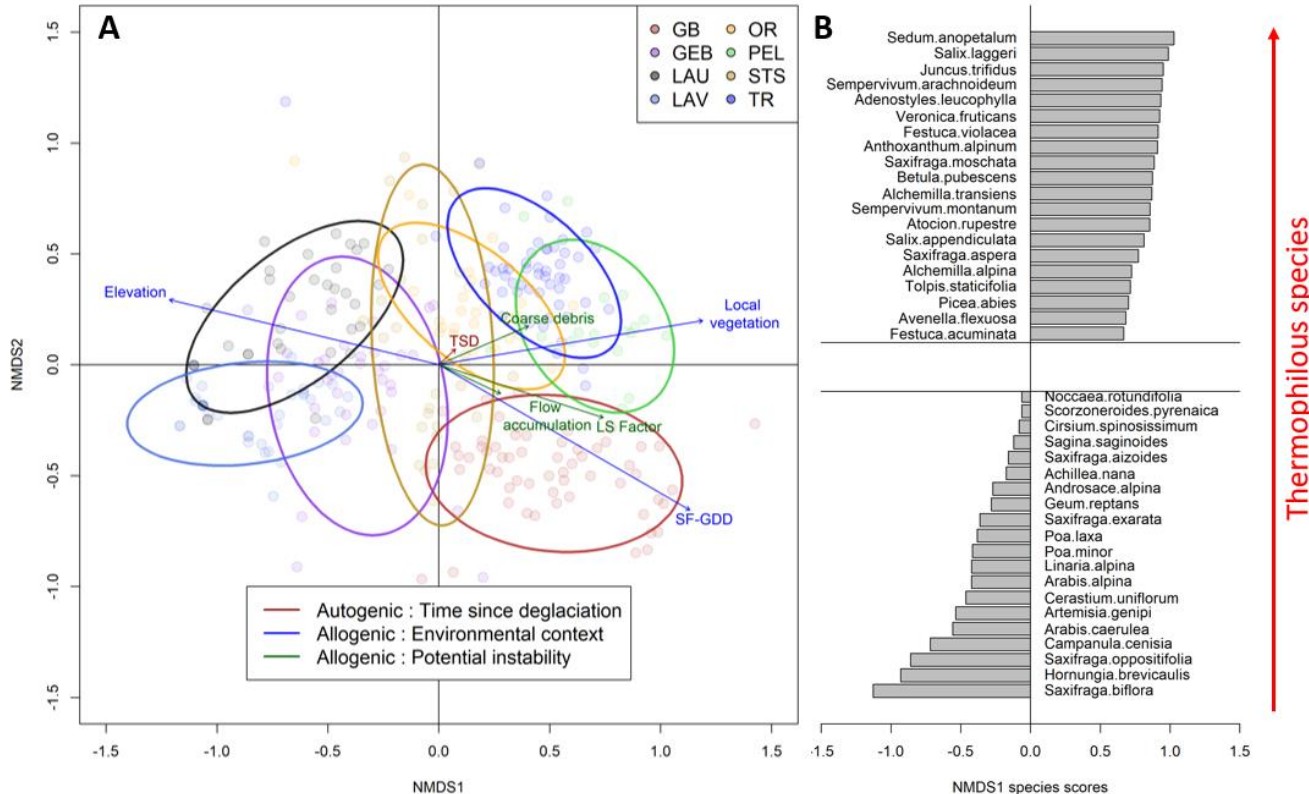

**Figure 8.** (A) Non-metric Multidimensional Scaling (NMDS) of floristic plots by species with vector fitting of three sets of variables described within the plot. Ellipsoids correspond to an interval of 0.8 the standard deviation. The analysis includes 297 plots from the 8 glacier forefields. (B) Species distribution along the NMDS axis 1 with the 20 species depicting the highest and lowest scores. According to the species distribution, the first axis corresponds to a thermal gradient with more thermophilous species as NMDS1 score increases.





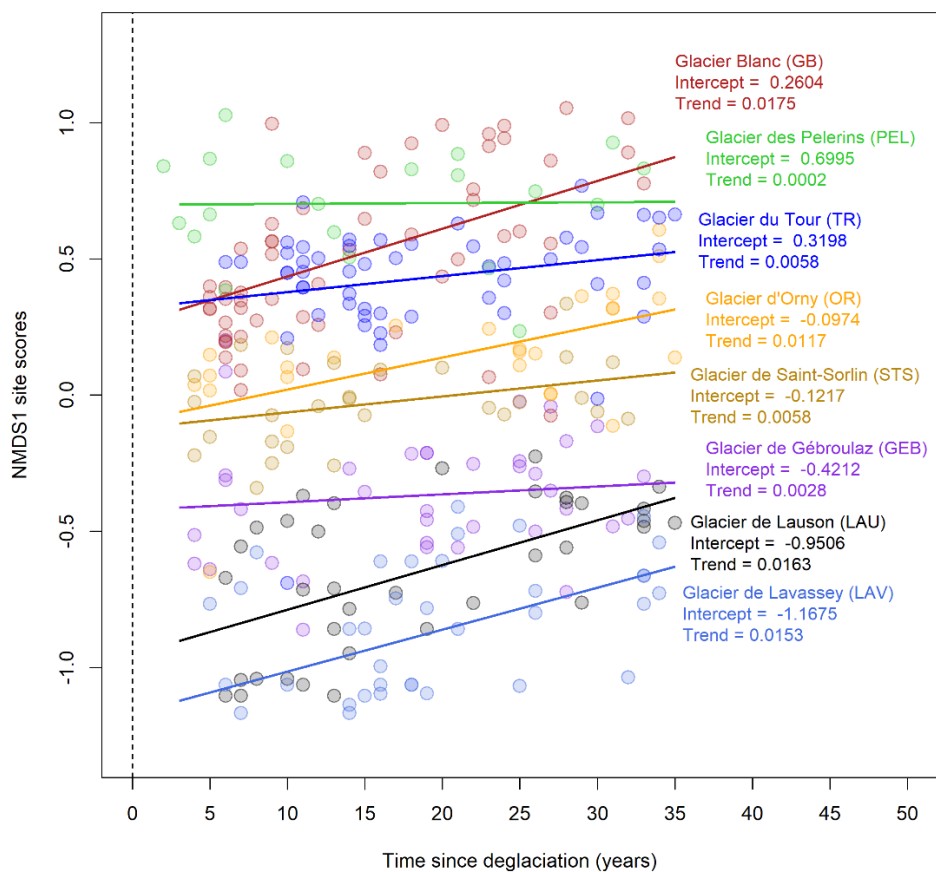

**Figure 9.** First axis scores of the NMDS according to time since deglaciation for the 8 glacier forefields. Intercept and trend based on linear models are shown for each glacier forefield.



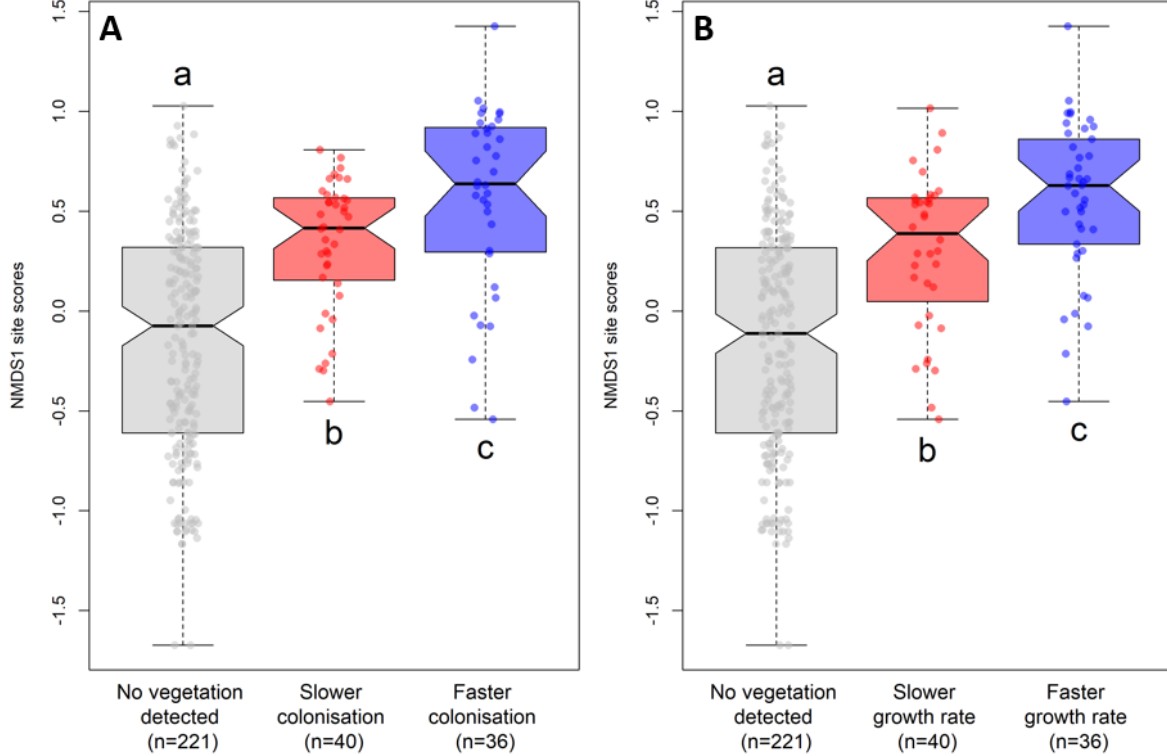

**Figure 10.** First axis scores of the NMDS according to the three classes of anomalies of (A) Time Lag and (B) Growth Rate
for an NDVI threshold of 0.075. Letters a, b and c indicate significant differences based on the Wilcoxon test (P-values <
0.05).





**Table 1.** R² values and significance level for NMDS ordination.

| Variables set | Variables | NMDS1 | NMDS2 | R² | P-value |
|---|---|---|---|---|---|
| Allogenic: Environmental context | SF-GDD (°C) | 0.8652 | -0.5013 | 0.7387 | 0.0001 |
| | Elevation (m. a.s.l.) | -0.9720 | 0.2349 | 0.6747 | 0.0001 |
| | Local vegetation (NDVI) | 0.9862 | 0.1654 | 0.6290 | 0.0001 |
| Allogenic: Potential instability | LS Factor (°) | 0.9376 | -0.3475 | 0.2621 | 0.0001 |
| | Coarse debris (cm) | 0.9177 | 0.3972 | 0.0836 | 0.0001 |
| | Flow accumulation | 0.9065 | -0.4220 | 0.0407 | 0.0028 |
| Autogenic | TSD (years) | 0.7265 | 0.6871 | 0.0046 | 0.5050 |