# Peer review of "Local environmental context drives heterogeneity of early succession dynamics in alpine glacier forefields"

_EGUsphere, 2022_

## Author Response (AR2)

Reviewers' comments are written in red while our responses are written in blue.

**REVIEWER 1 : THOMAS WAGNER**

Thank you Dr. Thomas Wagner for your precious and constructive comments on our work. We followed your recommendations which were mostly about the floristic plots analysis (NMDS) and implemented changes accordingly. We believe these changes drastically improved the quality, and foremost the clarity, of our work. We reviewed the entire manuscript to clarify the terminology used.

**GENERAL COMMENT**

**REVIEWER:** The well written manuscript provides valuable insights in the vegetation succession of glacier forefields. While the remote sensing part and the respective analysis is appropriate and the methods and encountered problems are comprehensively described, I have a number of critical comments regarding the evaluation and assessment of the vegetation.

**AUTHORS:** Thank you for your kind words.

**SPECIFIC COMMENTS**

1. *Succession dynamics in glacier forefields*

**REVIEWER:** In the discussion, the authors state that their results confirm that the time lag between deglaciation and plant establishment is dependent on the vegetation in the vicinity. However, I think that falls somewhat short, as the establishment of vegetation is generally also influenced by topographic factors and modulated by avalanches and debris flow etc. Larger and more frequent debris flows will clearly reduce (or reverse) recruitment even under equal seed pressure. Further higher elevations are not only accompanied with a less dense vegetation but also with generally lower establishment rates (due to climatic conditions). Although this does hardly weaken the results, this should be included into the discussion.

**AUTHORS:** You are right. The message that we are trying to convey is that 'vegetation in the vicinity' and energy availability (SF-GDD) are the main drivers of early succession dynamics. To do that, we compared their relative importance within a Random Forest framework, showing that these variables are good predictors. Conversely, we found that our 'potential geomorphological disturbances' variables were poor predictors of our two indicator variables (Time Lag and Growth Rate).

Nonetheless, as you stated, geomorphological disturbances have been shown to be of first importance in early succession in a large number of papers (we extensively discussed that point in the first paragraph of section 5.2). Accordingly, we know that our result appears counterintuitive. We believe that there are three reasons for this (the first two reasons are in part explained in section 5.2, while the third one came out as a new possibility thanks to your comment):

1. Our variables are poor predictors of actual geomorphological disturbances. Because we lack ground data on the history of geomorphological disturbances over our study site, we computed two variables (Flow accumulation and LS Factor). Both are intended to represent 'potential disturbances', following the idea that : "If a disturbance occurs, it would have the most impact where Flow accumulation and LS Factor are highest". However, in the case where no disturbance occurs, it is perfectly clear that these two variables would have zero predictive power. We found that our awareness about this issue was not clearly stated in our manuscript, hence we added a sentence in the last paragraph of section 5.2.

2. There is a scale issue regarding geomorphological disturbances. We described this in the last paragraph of section 5.2. Geomorphological disturbances tend to induce variability at the scale of glacier forefields, contributing to intra-forefield variability more than inter-forefield variability. Consequently, the importance of geomorphological variables and processes within glacier forefields might have been overshadowed by broader-scale drivers such as vegetation in the vicinity and energy availability.

3. Finally, in response to your next comment regarding the detection of changes from vegetation to unvegetated using the Landsat time series, it is probable that our approach is insensitive to these phenomena. In Figure 6C, we show that pixels identified as vegetated by our approach have on average 15% of plant cover in a 2x2 floristic plot, against 5% for pixels identified as unvegetated. It means that we do not directly detect "colonization" by vegetation but rather the surpassing of a certain threshold of "colonization" (Bayle et al. 2021). We discussed this in section 5.3. More importantly, it might mean that, because of the low sensitivity of our approach, we "*come after the geomorphological battle*", meaning that we only detect vegetation on surfaces that are stable enough, and thus less affected by geomorphological instability. This is a possibility that we did not initially discuss in our manuscript, so we added a paragraph in section 5.2. The fact that we also found poor predictive capabilities for geomorphological variables in the NMDS analysis might come from the known problem of undersampling in surfaces with known geomorphological instability due to dangerous access on the ground. We add a sentence in section 5.4 regarding this element. Line 434 : "*Finally, because of the low sensitivity of our remote sensing approach (discriminating vegetation cover around 10% only), it is possible that we only detect vegetation that develops on stable surfaces unaffected by geomorphological activities. In other words, we might only detect vegetation after the battle between substrate instability and colonisation by plants (Eichel et al. 2016), resulting in large underestimation of the importance of geomorphological activities.*"

We changed the section 5.2 title according to these new elements.

Line 428: "*We argue that the poor predictive capabilities of our geomorphological variables can be in part explained by the fact that those measures potential and not realised instability.*"

In addition, we found that we used the term "plant establishment" extensively while in fact, we should use "plant detection". We changed the terminology when appropriate.

**REVIEWER:** In this context, it would be also interesting to see the overall turnover of pixels from vegetation to unvegetated for each glacier (in addition to % vegetated, Fig. 5). This could give an idea of the debris flows and habitat loss typical for each glacier.

**AUTHORS:** Thank you for this suggestion. In fact, we looked at this possibility but did not find any significant trend. The main issue is that, as stated in our previous response, the sensibility of our remote sensing approach might be too limited to detect these events as we only detect surfaces with at least 10-15% of vegetation cover which might be only representative of stable surfaces. In addition, it is difficult in remote sensing based time series to decipher between noise and true events. For example, if after several years of vegetation detection the pixel is suddenly identified as unvegetated, then again vegetated after some years, it is impossible without proper ground control to attest that this is the result of disturbances rather than spectral related noise. In that manner, we did not explore these possibilities further. We add elements in the discussion regarding these elements in section 5.3.

Line 454: "*Nonetheless, because of coarse resolution and noises inherent to NDVI time series, our approach is unable to detect reverse trends with vegetated pixels changing to unvegetated pixels. For example, after several years of vegetation detection, a sudden drop in NDVI resulting in the pixel being identified as unvegetated could be interpreted as the result of disturbance or due to spectral-related noises, with no possibilities to decipher one from another. Improvement of our approach could be done by using proper ground control information regarding abrupt disturbances in glacier forefields. Also, systematic UAV flight on each glacier forefield could drastically improve the calibration of satellite remote sensing methods while providing relevant information on vegetation distribution and geomorphic activities at one time (Woellner & Wagner, 2019; Healy & Khan, 2022; Lang et al. 2021; Westoby et al. 2017).*"

2. *Plant assemblages and succession dynamics*

**REVIEWER:** This is the part I do have some problems with. The authors use about 15 plots per time period of deglaciation. Due to the different shape of the respective zones, for some of their sites the plots are far from each other while for other sites the plots lie quite close to each other. Hence spatial autocorrelation is likely, which should be considered when applying the test.

**AUTHORS:** See following responses for details about spatial autocorrelation.

**REVIEWER:** Another problem is the small plot size. 2x2m plots may be sufficient for lichens, but for higher vegetation they may be too small. Many rare species will be overlooked as they will only be detected using larger plots. Further, a minimum distance between plots may be associated with spatial autocorrelation, particularly when we assume that colonization is dependent on the seed pressure of the surrounding vegetation. Further, for example, while one Pinus mugo in a small plot might contribute considerable to the vegetation cover a small Saxifraga might be neglected. Hence correlating vegetation cover derived from remote sensing with a pixel representing ~30x30m and the vegetation survey on a 2x2m plot might be difficult.

**AUTHORS:** We agree that the 2x2m plot size for plant surveys represents certain methodological limitations, notably with respect to the 30x30m size of Landsat pixels. While developing the plant survey protocol, we did find that 17 random located sites with a quadrat size of 4m2 allowed for detection of 96% of vascular plant species present in a chronosequence band of the Glacier Blanc forefield (the "true" number of species was established using an exhaustive inventory approach). Accordingly, we are confident that from a floristic point of view, we were able to effectively capture plant communities present within the chronosequence of each forefield (Figure 1 below).

[Figure]

Figure 1. Area/Species richness curve from Parc National des Ecrins (PNE). C. Dentant (PNE), F. Anthelme (IRD).

While the intersection of floristic plots with Landsat pixels remains a delicate point, we did find consistent relationships between visually estimated plant cover in 4m2 quadrats and our classification of pixels being vegetated or non-vegetated in Fig. 6C. Furthermore, the results of our paper are primarily based on two separate methodological approaches (space-for-time floristic plots and real-time remote sensing), which converge to tell a similar story of plant colonization in our study sites (Fig. 7, Fig. 8). Aside from Fig. 6C, Figure 10 provides the only direct intersection of floristic and remote sensing indicators, and while the result is consistent with our hypotheses and the other findings of the paper, it is presented to deepen the discussion of the paper but does not constitute the core result of our work.

**REVIEWER:** For their NMDS the authors use all the plots of a site together, regardless of the period of deglaciation they represent. Hence, different successional stages are combined into one community. Inferences about different succession is consequently not possible, particularly if succession occurs at different rates for the individual glaciers. I would expect a separate consideration here or, if this is not possible only the communities after a certain time.

**AUTHORS:** You are right about the lack of coherence when including time since deglaciation within the NMDS framework as we use glacier forefields as clusters and not succession stage as it is most widely found. We removed the TDS variable from the NMDS framework and changed the manuscript accordingly. The most important result obtained from the

NMDS analysis is to be found in Figure 9 which shows the relation between NMDS1 and time since deglaciation. We show that the trend in NMDS1 score is almost always insufficient to surpass the starting point of the succession. We changed Figure 8 and Table 1 accordingly.

**REVIEWER:** Further, instead of a simple envfit, I would prefer to see a Mantel test for the respective environmental variables. Here the authors could also account for the spatial autocorrelation of the plots by simply including the plot coordinates (in UTM, meter) into the distance matrix. The results of the NMDS in terms of community composition (vegetation associated with the respective glacier forefield) should be discussed briefly.

**AUTHORS:** We acknowledge the importance of accounting for spatial autocorrelation when relating NMDS axes and variables. Instead of adding a new method (Mantel), we chose to improve the analysis by implementing a Generalized Least Square linear regression including or not an autocorrelated spatial error structure. This allowed us to select a best-fit model (based on AICc) for each variable. We modified Table 1 accordingly and revised Figure 8. The results section emphasizes the main changes brought by this more robust analysis.

Line 280: "*To account for spatial autocorrelation, we implemented two generalized least square (GLS) regression models for each variable: one including an autocorrelated error structure and one without. We used the spherical spatial correlation structure estimated by the function corSphere in the nlme package (Pinheiro and Bates, 2022). The best fitting model was selected using the Akaike Information Criterion corrected for small sample size (AICc). For each competing model, we estimated a pseudo-R-squared based on the regression of the variable on the fitted values. NMDS was performed using the metaMDS function of the vegan R package (Oksanen et al., 2020). GLS regression was fitted using the gls function of the nlme R package. Model evaluation was performed with the R package MuMIn (Barton, 2023).*"

Line 331: "*The GLS model including a spatial correlation error structure had better fit (lower AICc) for the three allogenic variables and for flow accumulation. The importance of the spatial structure was particularly strong for elevation with a pseudo-R2 of 0.74 without spatial autocorrelation to be compared to 0.23 with spatial autocorrelation. This indicated that the relationship between elevation and the floristic composition was mainly driven by the elevational differences between glacier forefields. For all other variables, we found no major changes in the relationships between NMDS axes and variables. *"

We added a discussion about the community composition in section 5.1. We argue that the identity of pioneer species and the structure of community composition, as well as the subsequent rate of growth following establishment (Fig. 10), are strongly influenced during the first decade of succession by energy availability and neighboring vegetation cover, and we refute a consistent "order of operations" hypothesis determining pioneer vs. non-pioneer species or functional groups in the context of glacial forefields in our study region.

Line 389: "*While previous studies have reported consistent pioneer plant species and functional groups giving way to later successional species in the context of glacier forefields (e.g. Shumann et al. 2016), our study of multiple glacier forefields indicates that the identity*

*of pioneer species varies highly from one site to another and depends strongly on local environmental context. Figure 9 shows that practically all of the plant communities encountered across the floristic gradient of the eight forefields have the potential to be pioneer species, given the wide range of initial starting points for initial community composition shown across NMDS scores. We emphasize that tree species such as Picea abies or Betula pubescens, or shrubs such as Salix laggeri, are just as capable of establishing quickly in the wake of glacier retreat as smaller stature forbs including Saxifraga or Poa spp. (Figure 8-9). Accordingly, we argue that the identity of pioneer species and the structure of community composition, as well as the subsequent rate of growth following establishment (Fig. 10), are strongly influenced during the first decades of succession by energy availability and nearby vegetation. "*

3.  *Methods general*

**REVIEWER:** For future research that does not rely on historical data, the use of UAV data should be discussed, as they provide high resolution just-in-time data (e.g. Woellner & Wagner, Healy & Kahn ...)

Woellner, R., & Wagner, T. C. (2019). Saving species, time and money: Application of unmanned aerial vehicles (UAVs) for monitoring of an endangered alpine river specialist in a small nature reserve. *Biological conservation*, *233*, 162-175.

Healy, S. M., & Khan, A. L. (2022). Mapping Glacier Ablation With a UAV in the North Cascades: A Structure-from-Motion Approach. *Frontiers in Remote Sensing*, 57.

**AUTHORS:** We added "*Also, systematic UAV flight on each glacier forefield could drastically improve the calibration of satellite remote sensing methods while providing relevant information on vegetation distribution and geomorphic activities at one time (Woellner & Wagner, 2019; Healy & Khan, 2022; Lang et al. 2021; Westoby et al. 2017).*" in section 5.3 of the discussion.

**TECHNICAL COMMENTS**

L114: heterogeneity in what context? please elaborate

We added "*(Time between deglaciation and plant colonization, and plant growth rate following colonization)*".

L145: please also provide the resolution of these data

We added *"(50 cm)*".

L253 and thereafter: Instead of "local vegetation" I would prefer the term "vegetation cover in the vicinity", to make clear that you do not look at the vegetation composition (which I expected, when I read the term first).

Thank you for this wording suggestion. You are right that it seems more clear using "vegetation cover" instead of "vegetation". We choose to use "Neighboring Vegetation Cover" instead of "Vegetation cover in the vicinity".

L292: Excluding species with less than 5 occurrences might be a problem, as this will exclude plots in early successional stages.

We opted to exclude species occurring with less than 5 occurrences throughout the floristic dataset in order to reduce stress in the NMDS analysis. This decision resulted in the removal of 24 species, which were concentrated in less than 1% of the plot, which given their rarity would have been anecdotal in structuring early succession dynamics and community composition. Still, we computed the NMDS with these rare species to analyze their importance. We found similar results with and without species with less than 5 occurrences.

L297: Please provide number of Iterations and dimensions for NMDS

We added: "*with 2 dimensions and 20 minimum random starts iterated two times*".

L299: see specific comments: better use Mantel tests instead of vector fitting

L350: Please consider that trees and shrubs have a generally higher seed rain and higher dispersal distances!

L379: I suppose this should be Fig. 3

Corrected.

L342ff: Please provide a table with all vegetation data (supplementary material). This is necessary to see the vegetation cover in context of the respective community.

We added the vegetation table in a Zenodo repository for visualization : https://zenodo.org/record/7698379. The link will be available in supplementary material.

L363: I would chose another term instead of heterogeneity here, as for me, heterogeneity suggests that the species composition differs

Caption Fig. 1: Please include to what chronosequence the colored lines relate to; I presume red is 0 Years, Blue 10 years …

Colored lines relate to order of deglaciation rather than the year of deglaciation. We changed the Figure 1 legend accordingly. Years of deglaciation can be found in Table S1.

L376ff: Describing the succession should include naming the relevant species and how community composition changes over time

Thank you Dr. Jana Eichel for your precious and constructive comments on our work. We provided discussion on your general comments and made changes in the manuscript accordingly. These changes clearly improved our manuscript.

**GENERAL COMMENT**

**REVIEWER:** The authors present an impressive and novel study quantifying the role of environmental heterogeneity on vegetation succession in glacier forelands. The study very nicely combines classical fields with novel remote sensing approaches, making it possible to compare vegetation succession across eight glacier forelands in the Alps. From the technical side, the authors dealt in my opinion very well with many difficulties arising in the study's context, such as assigning a continuous deglaciation age, problems with Landsat data quality etc. The manuscript is very well written and was a pleasure to read.

**AUTHORS:** Thank you for your kind words.

**REVIEWER:** I only have a few general comments and some specific comments (see below):

**REVIEWER:** When comparing your real-time remote sensing approach with a chronosequence approach, your time scale is limited to <40 years by the availability of Landsat data. So maybe the time since deglaciation becomes more meaningful on longer timescales (centuries) than you investigated and the chronosequence approach, despite all its large limitations, is still needed to investigate the complete vegetation succession since the end of the Little Ice Age. This needs to be mentioned and discussed.

**AUTHORS:** We agree with your comment. Accordingly, we made changes to the text regarding the interpretation of time since deglaciation in the NMDS, also in response Thomas Wagner's comments. We removed the TDS variable from the NMDS framework and changed the manuscript accordingly. The most important result obtained from the NMDS analysis is to be found in Figure 9, which shows the relation between NMDS1 and time since deglaciation. We show that the trend in NMDS1 score is almost always insufficient to surpass the starting point of the succession. We changed Figure 8 and Table 1 accordingly. In Figure 9, it is indeed possible that time since deglaciation will result in overstepping the determinism of the initial starting point over a longer time period (since LIA for example). We added "early" in the beginning of the discussion before "plant succession" to state more clearly that our results are only relevant for the beginning of the plant succession and that other mechanisms might be more important than local environmental context in later plant succession stages (as shown by Wojcik et al. 2021).

**REVIEWER:** In addition, I see the use of snow free-growing degree days somewhat critical, as you could only determine them for one season (2019). Over a period of nearly 40 years in a changing climate, I would expect that snow cover and growing degree days changed over and between the years, which would have affected vegetation succession. Thus, the reliability of this key indicator needs to be discussed. Looking at your NMDS and pairwise correlation results, it appears that SF-GDD is closely related to elevation anyway, so elevation of the glacier forelands might be a similarly important and more reliable factor for vegetation succession onset and dynamics.

**AUTHORS:** In this work, we used the snow free-growing degree days (SF-GDD) variable as a proxy of energy availability for plant growth by integrating the length of the growing period (informed by snow melt-out date from Sentinel-2) with modeled air temperature. In addition to elevation and aspect, it also integrates biogeographical elements which is key in our analysis as our study site spans 1° in latitude with difference in continentality. Accordingly, we consider SF-GDD to be a more meaningful ecological predictor compared to elevation. We agree that relying on only one year in a changing climate might appear limited but the SF-GDD gradient should be interpreted relatively rather than absolutely. Here, the absolute value of SF-GDD is not interpreted but only the relative position of pixels along the SF-GDD gradient.

As you mentioned, it also should be interpreted, in part, as elevation (which explains the strong negative correlation between these two variables). Elevation was kept in the NMDS for interpretation but Elevation, SF-GDD and Neighboring vegetation cover are three necessarily correlated variables.

**SPECIFIC COMMENTS**

l. 57: Space-for-time approaches: This method does not necessarily rely on the position of plant surveys to estimate time since deglaciation, but can also be done using known terrain ages from glacier stages, maps, dating etc. (cf. Matthews, 1992, "The ecology of recently deglaciated terrain"). Please revise.

We adapted the text to be more generalist and accurate.

l. 213: "Thus, we applied the same method as in Bayle et al. (2022)". Please add details which method exactly and for which purpose.

We clarified the method used.

l. 222: Add "The" before "complete workflow"

Done

l. 237: add "-" after intra

Done

l. 315 ff: If I understand your interpretation of Figure 4 correctly, you are looking at when the mean NDVImax per terrain age class crosses the NDVI threshold? Because the error bars

seem to imply that some pixels already crossed that threshold earlier than stated here in the text? If this is the case, please clarify in the text that you are talking about mean NDVImax.

You are right. The time specified in the text corresponds to the number of years it takes for the average NDVImax to reach the threshold. We clarified this in the text.

I. 336 "Floristic plots are representative of glacier forefields vegetation dynamics". Wouldn't it be the other way around that your Landsat detection matches what is happening at the ground in the plots?

Yes, in a way. But here our thinking is the following: floristic plots on the ground were done with limitation due to field sampling, which could have hindered the representativity of the whole forefield (because more limited or disturbed vegetation tends to be less represented in field sampling because it might be less accessible). With Landsat, we consider that we have information on the entire forefield (even if it is obviously degraded information compared to field sampling) and thus that we are able to capture the "true" heterogeneity of vegetation dynamics within study areas. We showed that for our two remote sensing indicators (TL and GR), variance was similar when considering the entire margin or only pixels overlapping field sampling, meaning that our field sampling captured the overall heterogeneity of the margin's vegetation. We acknowledge however that if we considered our field sampling to be representative, and that remote sensing is not necessarily spatially representative, we could argue that we are indeed demonstrating that Landsat detection matches what is happening at the ground in the plots. We modified the section title to more clear.

Line 315: "*Agreement between field and Landsat observations of plant cover*"

I. 395: "dynamic" in which sense? Vegetation colonization? Please specify.

We specified "(in term of vegetation colonization rate)"

I. 451: "periglacial"

Done

I. 460: "that" instead of "the"

Done

I. 470: I would not term natural disturbances such as geomorphic processes in glacier forelands "erratic". They do follow a certain pattern in time during paraglacial adjustment (e.g. Ballantyne et al., 2002; Eichel et al. 2018) and geomorphic processes can have a certain magnitude-frequency distribution. So those geomorphic disturbances are not erratic but can follow certain patterns.

We changed "erratic" by "varying".

Figure 3: Which data is exactly shown ? Changes of all pixels in one glacier foreland? Changes of one pixel at the site of the coordinates given? Please provide more information in the figure caption.

We corrected the Figure 3 caption which had several wrong pieces of information. We clarified that panel A was for only 1 pixel in Glacier Blanc (corresponding to the coordinates). Panel B and C represent data for the 8 margins.

Fig. 10: Missing reference to Fig. 10 in results section.

Corrected

Table 1: Coarse debris is given as an indicator for instability – is this correct or would it more be an indicator for stability?

Terminology: different terms are used to refer to the interactions between ecologic and geomorphic processes, e.g. bio-geomorphic and eco-geomorphological. I would suggest you choose one term to use throughout the manuscript, most commonly used is in my opinion biogeomorphic (ecogeomorphology was first defined for fluvial systems).

Done